# Trapped charge-driven degradation of perovskite solar cells

Namyoung Ahn[1,2,*], Kwisung Kwak[1,2,*], Min Seok Jang[3], Heetae Yoon[1,2], Byung Yang Lee[4], Jong-Kwon Lee[1], Peter V. Pikhitsa[1], Junseop Byun[1,2] & Mansoo Choi[1,2]

Perovskite solar cells have shown unprecedent performance increase up to 22% efficiency. However, their photovoltaic performance has shown fast deterioration under light illumination in the presence of humid air even with encapulation. The stability of perovskite materials has been unsolved and its mechanism has been elusive. Here we uncover a mechanism for irreversible degradation of perovskite materials in which trapped charges, regardless of the polarity, play a decisive role. An experimental setup using different polarity ions revealed that the moisture-induced irreversible dissociation of perovskite materials is triggered by charges trapped along grain boundaries. We also identified the synergetic effect of oxygen on the process of moisture-induced degradation. The deprotonation of organic cations by trapped charge-induced local electric field would be attributed to the initiation of irreversible decomposition.

[1] Global Frontier Center for Multiscale Energy Systems, Seoul National University, Seoul 08826, Korea. [2] Department of Mechanical and Aerospace Engineering, Seoul National University, Shinlimdong, Seoul 08826, Korea. [3] School of Electrical Engineering, Korea Advanced Institute of Science and Technology, Daejeon 34141, Korea. [4] Department of Mechanical Engineering, Korea University, Seoul 02841, Korea. * These authors contributed equally to this work. Correspondence and requests for materials should be addressed to M.C. (email: mchoi@snu.ac.kr).

Metal halide perovskite solar cells[1-3] have shown unprecedent performance increase up to 22% efficiency[4] and are now considered not only as a low-cost alternative to commercialized solar cells[5,6] but also as a functional cell with flexible applications[7,8]. However, their long-term stability has not been solved yet and this issue is the most pressing problem for commercialization[9]. It is well known that perovskite materials are vulnerable to the exposure of humidity and light[10-12]. Although many efforts to encapsulate the devices for preventing direct contact to humidity have been attempted, it was not successful to obtain long-term stability comparable to commercial photovoltaic devices[13]. Various factors that can affect the stability have been investigated from the viewpoints of chemical structure[14], electrical stress[15], hydrated states[10,11,16] and heat[17]. However, the degradation mechanism is still unclear how a fully fabricated device deteriorates rapidly, even though its perovskite layer is tightly covered by the hole transport material (HTM) and the back electrode. It is also elusive why light soaking causes irreversible degradation of perovskite materials in the presence of moisture, whereas the moisture in the dark condition only induces reversible hydration of perovskite materials[11,16].

The device structure and the choice of charge extraction materials also influence the stability[18]. The use of inorganic charge extraction layers was reported to enhance stability[13,19]. Other studies have focused on eliminating the pathway of water vapour infiltration into the perovskite film by coating carbon-based materials, polymers and hydrophobic materials on the top surface of the perovskite film[12,20]. These approaches provide the device lifespan longer than the case without these materials, but not long enough to ensure long-term stability. Such approaches also occasionally sacrifice the photovoltaic performance. Especially, the devices employing titanium dioxide ($TiO_2$) as electron transport layer (ETL) are rapidly degraded under light soaking[21], even though they are highly efficient in energy conversion and exhibit the world's best efficiency.

In the present study, we demonstrate that the charges trapped at the interface between perovskite and charge extraction materials are responsible for the irreversible degradation due to moisture. The elusive experimental puzzles both on the occurrence of degradation beginning from different side depending on different charge extraction material and the role of light soaking for irreversible degradation will be clearly explained from the present concept of the trapped charges later. To dig into this charge-driven degradation mechanism, we investigated controlled stability experiments both for commonly used $CH_3NH_3PbI_3$ ($MAPbI_3$), which is known to form structurally distorted tetragonal crystals, and a mixed perovskite material having more enhanced structural stability. The crystal structure of perovskite can become more stable by increasing the tolerance factor close to unity by incorporating other organic cation and halide anion with different ion sizes[22,23] (see Supplementary Fig. 1). Addition of formamidinium (FA) cation and bromide (Br) anion could not only structurally stabilize the perovskite materials by increasing tolerance factor towards one for inducing cubic crystals but also enhance their photovoltaic performance by broadening absorption spectrum[5,6]. Although several studies on mixed cation and/or halide anion system of $MA_xFA_{1-x}PbI_yBr_{3-y}$ have been suggested[5,6,24], they focused on the performance aspect, not the stability aspect. Therefore, an effort is required to develop new composition perovskite, ensuring both stability and high performance. In this work, we developed a mixed $MA_{0.6}FA_{0.4}PbI_{2.9}Br_{0.1}$ perovskite, ensuring both high performance and stability via Lewis base adduct method[25] (see Methods section, Supplementary Figs 1–4 and Supplementary Note 1). Later, it will be shown that this mixed perovskite still degrades, although its degradation speed is slower than the case of conventional $MAPbI_3$ and the irreversible degradation of both perovskites is triggered by trapped charges.

## Results

**Performance and stability comparison depending on ETLs.** First, we examined how solar cell degradation behaviour becomes different depending on different charge extraction layers, for example, $C_{60}$ and compact $TiO_2$ (see the detailed process for solar cell fabrication in the Methods section). Figure 1a,b show $J$–$V$ curves for $MA_{0.6}FA_{0.4}PbI_{2.9}Br_{0.1}$ perovskite on the $C_{60}$ ETL (Fig. 1a) and compact $TiO_2$ layer (Fig. 1b), respectively. Figure 1a shows $J$–$V$ curve for our mixed perovskite on 35 nm-thick, dense $C_{60}$ layer deposited by thermal evaporation, which demonstrates hysteresis-less performance of the best power conversion efficiency ($PCE$) of 20.2%. The best $PCE$ value was averaged from the $J$–$V$ curves of forward and reverse scan, which is in agreement with 20.2% of steady-state efficiency shown in Supplementary Fig. 4b. The integrated $J$sc estimated from external quantum efficiency (EQE) was also well-matched with the measured $J$sc as shown in Supplementary Fig. 4c. Histograms of the short-circuit current ($J_{sc}$), the open-circuit voltage ($V_{oc}$), the fill factor and the efficiency of 47 cells are shown in Supplementary Fig. 4d–g. The photovoltaic characteristics of these cells were highly reproducible with a small s.d. and the average values are $J_{sc} = 24.34$ mA cm$^{-2}$, $V_{oc} = 1.058$ V, fill factor = 0.743 and $PCE = 19.12\%$, respectively. This would be the best performance of low-temperature processed perovskite solar cells without using mesoporous $TiO_2$. On the other hand, the case coated on 40 nm-thick compact $TiO_2$ showed a large hysteresis with $PCE$ of 16.9% on the reverse scan and 9.0% on the forward scan, which is consistent with previous studies that also showed a large hysteresis for compact $TiO_2$-based devices[26].

As shown in Fig. 1c,d, non-encapsulated $C_{60}$-based cell shows much more stable performance under one sun illumination but still degrades, whereas compact $TiO_2$-based non-encapsulated cell completely died only after 6 h. To examine the detailed evolution of degradation, we investigated how the cross-sectional morphology of the $C_{60}$- and $TiO_2$-based devices would evolve under illumination via the focused ion beam-assisted scanning electron microscope (SEM) images shown in Fig. 2. Consistent with the $PCE$ measurement results, the SEM images clearly confirm that the $C_{60}$-based devices are much slowly degraded compared with the $TiO_2$-based cells. Strikingly, they showed different degradation patterns, namely different side of degradation beginning where the degradation is initiated depending on different ETLs. Such degradation pattern is the same for conventional $MAPbI_3$ perovskite (see Supplementary Fig. 5). As the reactants that can decompose perovskite materials could infiltrate from the thin metal electrode rather than from the thick indium tin oxide-coated (ITO) glass, it would be expected that the degradation should be initiated at the interface closer to the thin Au metal electrode. However, the perovskite films of $TiO_2$-based devices began to be decomposed at the interface adjacent to the compact $TiO_2$ layer near fluorine doped tin oxide (FTO) glass (as shown in Fig. 2b). Although these observations were attributed to ultraviolet light-induced photocatalytic effect of $TiO_2$ layer according to the previous report[27], we confirmed that the same degradation pattern happened in the $TiO_2$- and $C_{60}$-based devices aged even under ultraviolet-filtered light illumination (see Supplementary Fig. 6). For those of $C_{60}$-based devices, the decomposition began from the interface adjacent to HTL near Au metal electrode opposite to the case of $TiO_2$-based devices. As the two types of devices have identical structure, except for the ETL, $C_{60}$ or $TiO_2$/Perovskite/Spiro-MeOTAD/Au, these different degradation characteristics indicate that charge extraction may play an important role where moisture-driven decomposition of

perovskite material begins. Second, the TiO$_2$-based devices suffer from severe hysteresis, whereas the C$_{60}$-based devices do not. Considering that the origin of hysteresis is known as capacitive current[26], trapped charge[28] and unbalanced charge injection[29], many electrons may be accumulated near the ETL in the TiO$_2$-based devices, whereas the C$_{60}$-based devices hardly do. From the observation on the degradation of the TiO$_2$-based devices that begins from the interface contacting TiO$_2$ layer where many charges could be trapped, it is reasonable to suspect that trapped charges at the interface between perovskite and charge extraction layer would be responsible for initiating the moisture-related decomposition. Fast extraction of electrons through C$_{60}$ would hardly accumulate negative charges at the interface between perovskite and C$_{60}$, but hole extraction through Spiro-MeOTAD could be slower than the rate of electron extraction in the C$_{60}$-based cell[30,31]. This could result in positive charge trapping at the interface between perovskite and hole extraction layer, which could be the cause why the degradation begins from the interface between perovskite and Spiro-MeOTAD for C$_{60}$-based cells (see Fig. 2a). These results demonstrating the degradation beginning from opposite side for different charge extraction layers gave us a clue about the trapped charge-driven degradation regardless of polarity.

**Hydration and light-induced degradation of perovskite materials.** Another intriguing experimental observation is the light soaking in the presence of moisture, which consistently showed

irreversible degradation of perovskite in previous works[9,11,12], whereas in the dark condition moisture introduction only formed reversible hydrates of perovskites, for example, $CH_3NH_3PbI_3 \bullet H_2O$ or $(CH_3NH_3)_4PbI_6 \bullet 2H_2O$ (refs 11,16). The reason has not been elucidated yet, although Christians et al.[11] suggested that organic cation could become less tightly bound to $PbI_6^{4-}$ octahedra after light soaking. In the present study, along with the scenario of the above mentioned trapped charges that could trigger irreversible degradation, the charge generation under light soaking and subsequent trapping on the surface of perovskite is suspected to initiate the moisture-induced irreversible degradation under light illumination. To confirm the irreversible degradation under light soaking, we also investigated the experiments under light soaking or not in the presence of moisture. Figure 3a–f showed the degradation behaviour of MAPbI$_3$ and our mixed MA$_{0.6}$FA$_{0.4}$PbI$_{2.9}$Br$_{0.1}$, respectively, for 2 days in the dark condition with relative humidity (RH) 90%. Absorption spectra measurements show that the original MAPbI$_3$ (black curve) became hydrated (red curve) after 2 days and then dehydrated reversibly via N$_2$ drying, which is consistent with previous studies[11,16]. On the other hand, the absorption spectra of our mixed MA$_{0.6}$FA$_{0.4}$PbI$_{2.9}$Br$_{0.1}$ perovskite were hardly changed with the same condition (see Fig. 3d) and X-ray diffraction (XRD) patterns were the same after 2 days (see Supplementary Fig. 7). This indicates our mixed composition perovskite would be more resistible to become hydrated than distorted tetragonal perovskite MAPbI$_3$. It is likely to be that

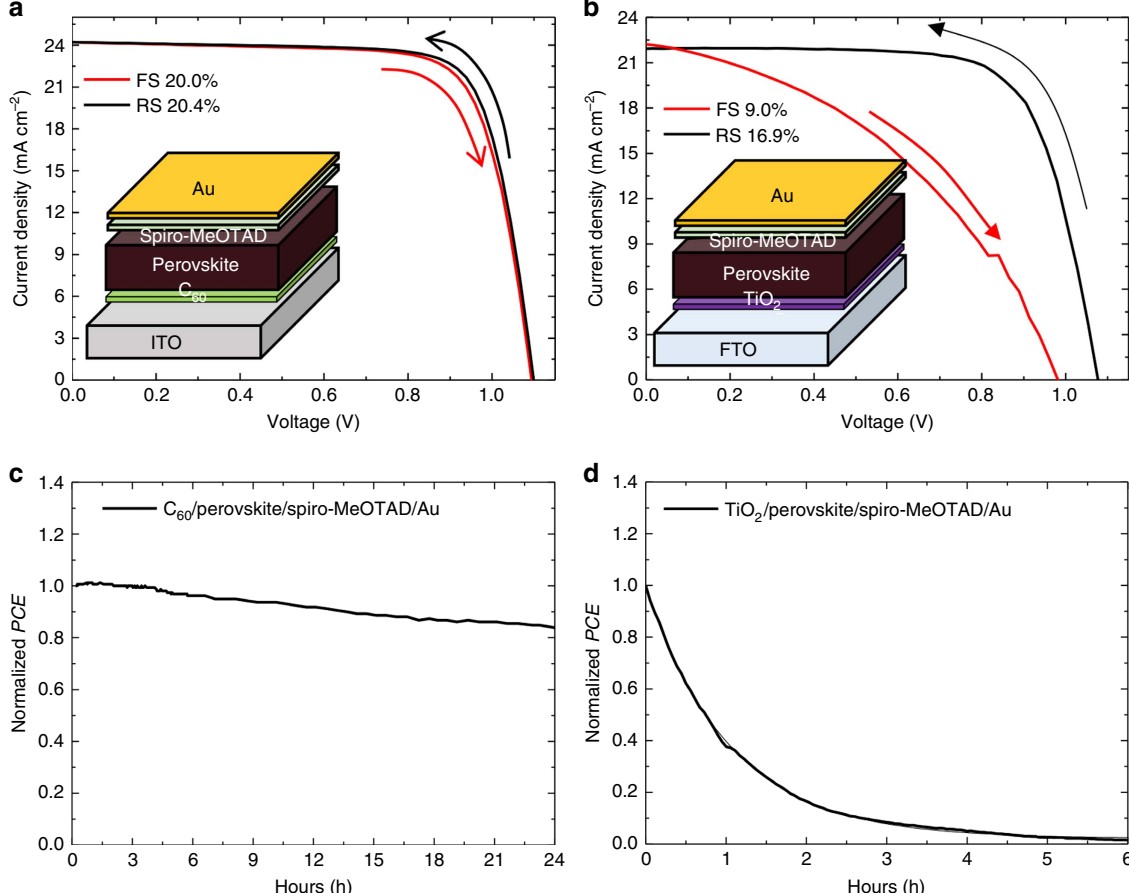

**Figure 1 | Device performance and stability depending on different ETLs.** J–V curves of (**a**) ITO/C$_{60}$ (35 nm) /MA$_{0.6}$FA$_{0.4}$PbI$_{2.9}$Br$_{0.1}$ (500 nm)/Spiro-MeOTAD (250 nm)/Au (50 nm) and (**b**) FTO/TiO$_2$ (40 nm)/MA$_{0.6}$FA$_{0.4}$PbI$_{2.9}$Br$_{0.1}$ (500 nm)/Spiro-MeOTAD (250 nm)/Au (50 nm) measured in the reverse (black) and forward (red) scans with a 200 ms sweep delay. (**c,d**) Time evolution of the normalized PCE measured under one sun illumination in ambient conditions (RH = 30%) of the (**c**) C$_{60}$- and (**d**) TiO$_2$-based devices.

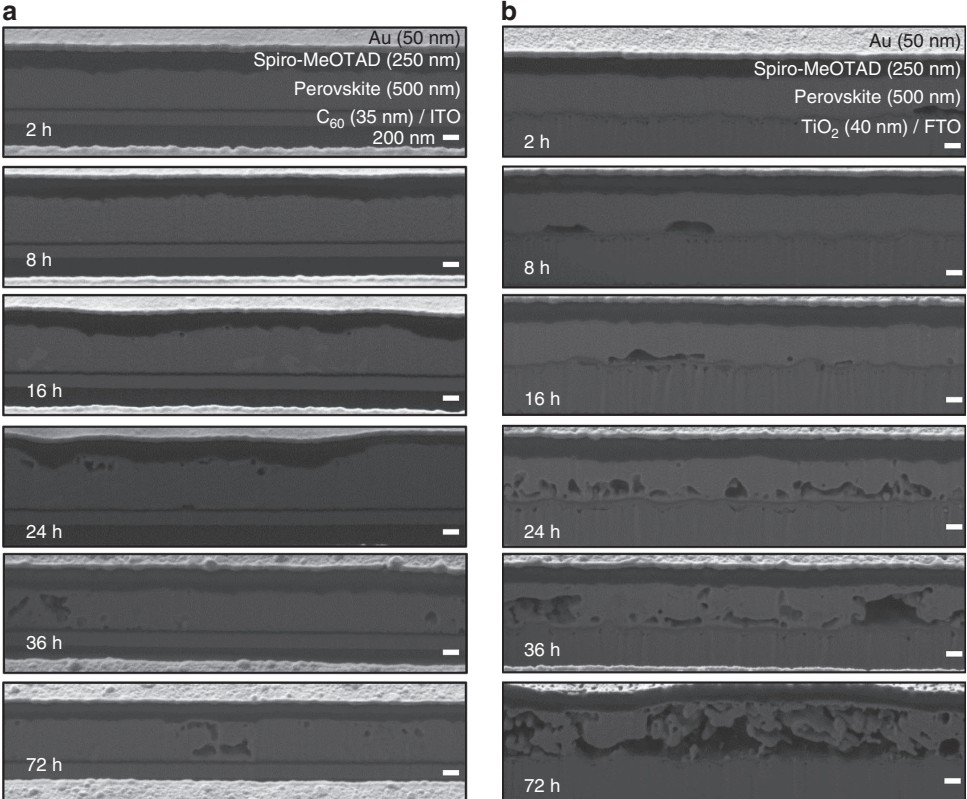

**Figure 2 | Degradation patterns depending on different ETLs. (a,b)** Time evolution of the focused ion beam-assisted SEM (FIB-SEM) cross-sectional images of the (**a**) $C_{60}$- and (**b**) $TiO_2$-based devices aged for 72 h under light illumination in ambient conditions. Scale bars, 200 nm.

water molecules could penetrate more easily into the distorted tetragonal $MAPbI_3$ than into the more compact cubic crystal structure of $MA_{0.6}FA_{0.4}PbI_{2.9}Br_{0.1}$ (ref. 23). A slight change of the absorption spectra shown in Fig. 3d indicates a slow hydration could still happen to our mixed perovskite under 90% RH. However, both perovskites showed irreversible degradation rapidly under light soaking even at low *RH* 20% (see Fig. 3g–l), which is also consistent with previous report[11,12,21]. It is interesting to note that under light soaking, our mixed perovskite degrades more slowly than the conventional $MAPbI_3$. As mentioned earlier, perovskite-absorbing light can generate and store charges due to its capacitive property[32] that may be trapped on the grain boundaries (we will show trapped charges along grain boundaries after light soaking later). It could be hypothesized that these trapped charges generated under light soaking would be responsible for irreversible degradation, which is in line with the aforementioned hypothesis of trapped charge-driven degradation explaining the initiation of degradation on different side, depending on different charge extraction layers.

**Trapped charge-driven degradation.** To prove this compelling hypothesis of trapped charge-driven degradation, we have configured an experimental setup employing an ion generator by corona discharge and a stainless chamber that blocks all incident light from outside as shown in Fig. 4a. Detailed descriptions of the ion generation via corona discharge method and ion deposition can be found in the refs 33–35. The air inside the chamber is isolated from the outside and controlled by two gas inlets that are connected to the independent gas sources (Gas 1 and Gas 2; see the Supplementary Note 2). Gas 1 is ionized by applying a high voltage to the pin of the corona chamber, delivered to the deposition chamber by gas flow and deposited on the perovskite film placed at the bottom of the deposition

chamber. Gas 2 passes through a water bubbler to regulate the humidity level in the deposition chamber. We measured the time evolution of the absorption spectra as the perovskite films were gradually degraded in the deposition chamber. Gas 1 was chosen as nitrogen or hydrogen for generating positive or negative ions, for being used as different polarity charges trapped on the surface of perovskite, respectively, whereas Gas 2 was nitrogen (see the Supplementary Note 2). First, we needed to check that $N_2$-positive ions and $H_2$-negative ions themselves do not affect the degradation without moisture (see Supplementary Fig. 8). Next, we examined the degradation behaviour in the presence of moisture when charged ions deposited on the surface of perovskite. When the positively charged $N_2$ ions were deposited and the RH in the chamber was held at 40%, the perovskite film rapidly decays as shown in Fig. 4b. The deposition of negatively charged $H_2$ ions also showed the similar irreversible degradation behaviour under the same moisture level in Fig. 4c. Although Fig. 4c for negative charges appears to cause slower degradation than the case shown in Fig. 4b for positive charges, this could not tell which polarity charges affect more adversely on the degradation, as ion generation for different polarity in our experiment is different (see the Supplementary Note 2). It is noteworthy that, on the other hand, without depositing charges, the absorption spectra and XRD patterns of the perovskite film were hardly changed for 2 days even under 90% humidity as was shown in Fig. 3d and Supplementary Fig. 7. Similarly, at the presence of only charges without moisture, the degradation did not occur at all as shown in Supplementary Fig. 8. This suggests that the irreversible degradation of perovskite materials only take place when both moisture and charges exist simultaneously. Moreover, the structurally distorted (conventional) $MAPbI_3$ film was degraded more quickly than the mixed stable $MA_{0.6}FA_{0.4}PbI_{2.9}Br_{0.1}$ under the same moisture and ion deposition level (Supplementary

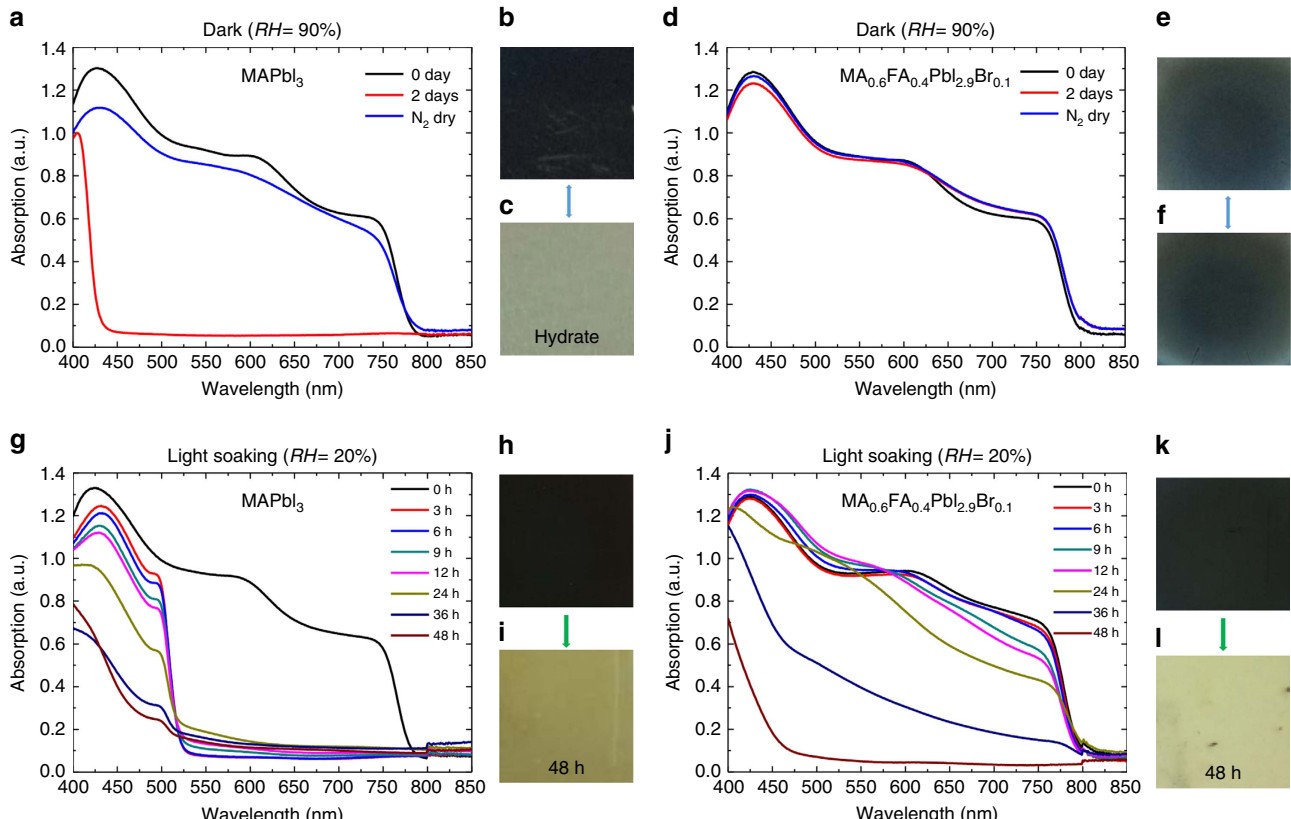

**Figure 3 | Hydration and light-induced degradation of perovskite materials.** (**a**) Absorption spectra and pictures of the MAPbI₃ perovskite film (**b**) before and (**c**) after ageing for 2 days under dark conditions at 90% RH. MAPbI₃ perovskites were transformed into transparent hydrated states after 2 days. (**d**) Absorption spectra and pictures of the MA₀.₆FA₀.₄PbI₂.₉Br₀.₁ perovskite film (**e**) before and (**f**) after ageing under the same condition. (**g**) Time evolution of absorption spectra and pictures of MAPbI₃ (**h**) before and (**i**) after degradation under light soaking at 20% RH. (**j**) Time evolution of absorption spectra and pictures of MA₀.₆FA₀.₄PbI₂.₉Br₀.₁ (**k**) before and (**l**) after degradation under the same condition.

Fig. 9). Based on these observations, the degradation mechanism could be thought of a two-step process: the formation of hydrated perovskite by humidity and the irreversible decomposition by trapped charges. The first step of the formation of hydrated perovskite was already reported by several groups[10,11,16]. Here we suggest that local electric field caused by trapped charge could distort electrostatically the structure of hydrated perovskite in which octahedral $PbX_6^{4-}$ interacts with both organic cation and $H_2O$, and trigger the initiation of irreversible decomposition of perovskite (will discuss more complete scenario in the Discussion section). Leijtens et al.[9,15] found the irreversible degradation near the gold electrodes coated on perovskite film by applying a weak external field of $600\,V\,cm^{-1}$ in the presence of moisture and attributed it to the ion movement through electric field. As an electric field was applied between two electrodes touching perovskite film, electric current could flow and there was a possibility of charge trapping underneath the electrode, which might have played a role for degradation. To isolate the effect of pure external electric field, we examined the degradation of perovskite materials by applying non-contact electric field, which was given by two floating electrodes; one electrode is in air above perovskite film coated on ITO glass and the other electrode exists beneath the glass. We found no degradation up to $12\,kV\,cm^{-1}$ under 90% RH (see Supplementary Fig. 10). It is noteworthy that this field will be dropped across the air gap and, therefore, the real field inside perovskite film should be different from the given field and the perovskite film might be uniformly polarized by one-directional strong **E**-field, because perovskite materials have a high dielectric constant[36,37]. Therefore, further study should be needed to completely understand the effect of pure external electric field. It

is noted that the differences between the fields due to the trapped charges and the external field lie in the point-like character of the trapped charges, which produce locally huge and irregular fields. The huge and irregular fields formed by charges trapped along grain boundaries could help the process of deprotonation[27,38] (will discuss in the Discussion section).

Next we investigated how trapped charge could decompose perovskite material in time by examining morphology evolution via SEM analysis. As shown in Fig. 5e–l (tilted top-view and cross-sectional images) and Supplementary Fig. 11 (top-view images), the degradation is initiated from the grain boundaries. As the reaction continues, the colour of the film turns into yellow (Fig. 5a–d), indicating that the perovskite is irreversibly decomposed to $PbI_2$ (see XRD patterns after degradation in Supplementary Fig. 12). It is interesting to dig into why degradation occurs from grain boundaries in line with our trapped charge mechanism. To check the distribution of trapped charges on the surface of perovskite after uniform ion deposition, we measured Kelvin probe force microscopy (KPFM) of untreated perovskite and ion-treated perovskite films (see the Supplementary Note 3). Figure 5m,n shows topology and surface potential distribution of the perovskite surface on which positive $N_2$ ions were uniformly showered. Striking coincidence between two images is the evidence that charges are preferentially trapped along grain boundaries. Overlapped image of topology and potential distribution shown in Supplementary Fig. 13 clearly demonstrates charges are trapped along grain boundaries even though ions are showered uniformly. For untreated sample, there is no correlation between topology and potential distribution (see Supplementary Fig. 13). With this charge trapping along

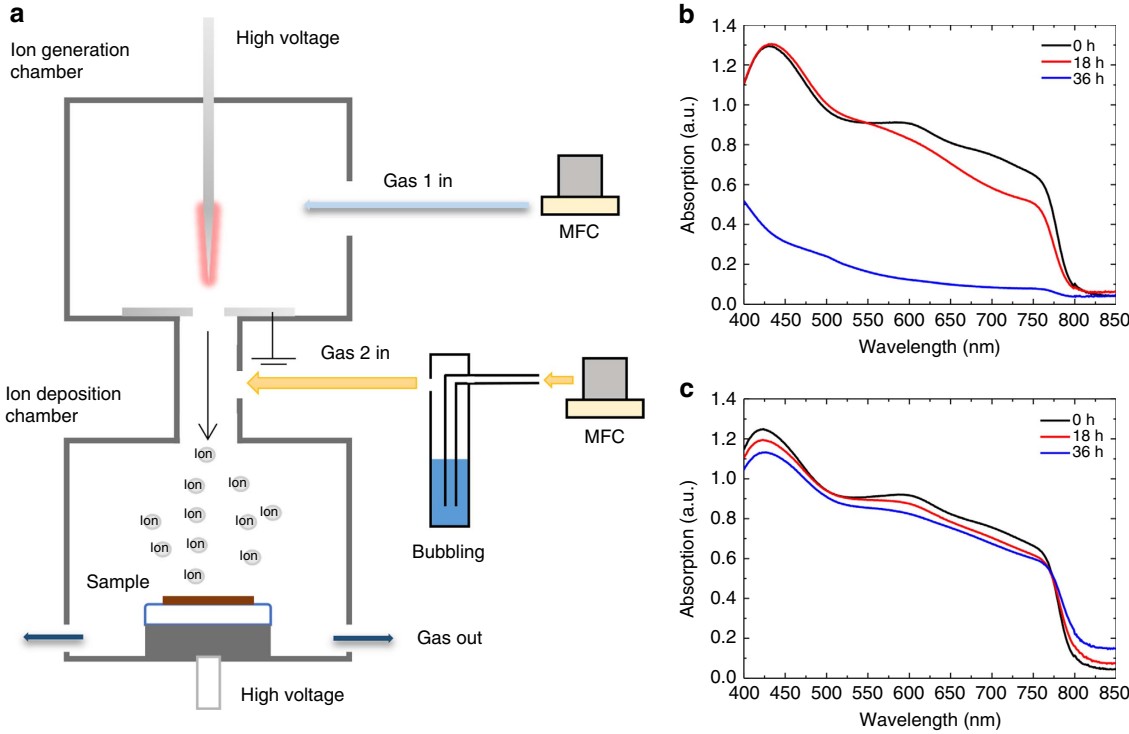

**Figure 4 | Experiments for trapped charge-driven degradation.** (**a**) Experimental setup of corona discharge for ion generation, bubbling system for humidification and SUS chamber for ion deposition and blocking light. (**b,c**) Absorption spectra of the perovskite film measured at an interval of 18 h during deposition of (**b**) positive nitrogen ions and (**c**) negative hydrogen ions at 40% RH.

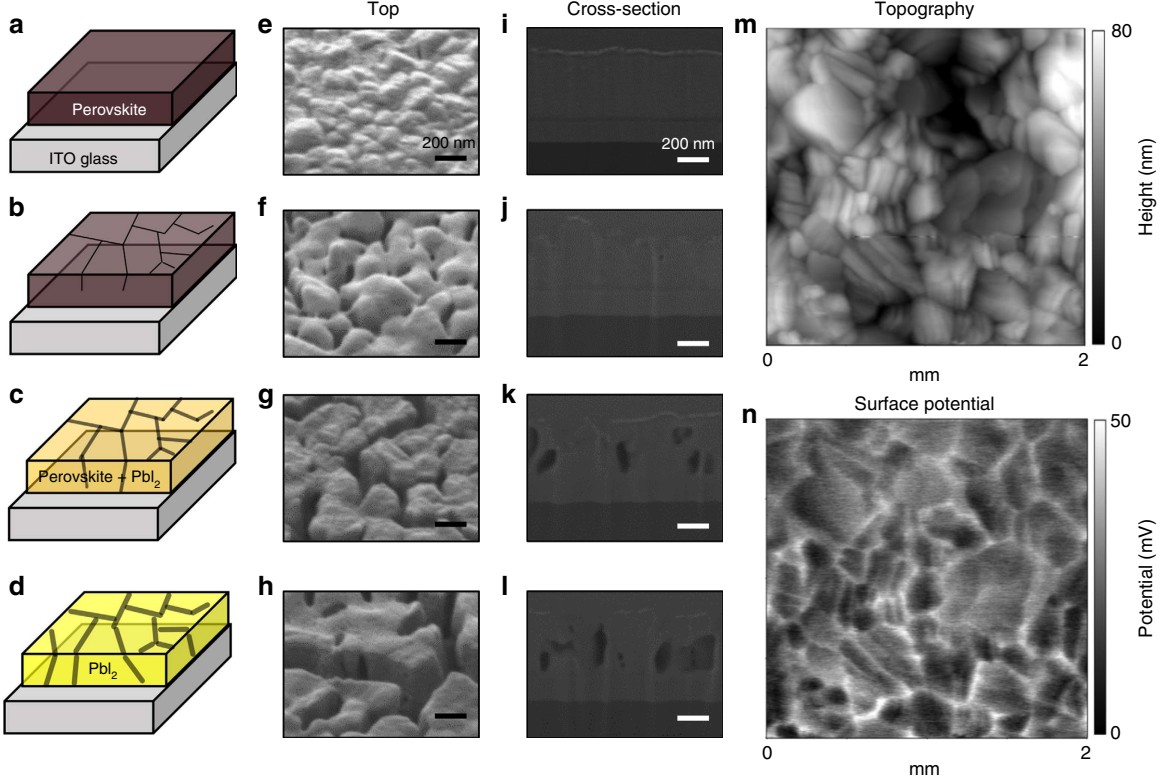

**Figure 5 | SEM and KPFM measurements.** (**a–d**) Schematic illustration of perovskite degradation processes (left), (**e–h**) top-view (middle) and (**i–l**) cross-sectional (right) SEM images of perovskite layers (**a,e,i**) before, (**b,f,j**) after 6 h, (**c,g,k**) 12 h and (**d,h,l**) 18 h by ion deposition in humidified nitrogen. The colour change from dark brown to yellow in **a–d** represents the gradual degradation process. Black lines and their widths in **a–d** represent grain boundaries and degradation extent, respectively. Scale bars, 200 nm. (**m**) Topography and (**n**) surface potential profile of $MA_{0.6}FA_{0.4}PbI_{2.9}Br_{0.1}$ film obtained from KPFM measurements after deposition of $N_2$-positive ions.

grain boundaries, experimentally observed degradation pattern following grain boundaries and the fact that the irreversible degradation occurs only when moisture and charges exist together are the evidences that trapped charges would be responsible for the initiation of irreversible degradation of perovskite materials. It is now apparent that grain boundaries are the most vulnerable sites for degradation, because they provide charge accumulation sites as well as infiltration pathway of water vapour[9]. Successful enhancement of stability using high-mobility inorganic charge extraction layers supports the present idea[13,19].

To further investigate the possibility that the above mentioned intriguing experimental observation of irreversible degradation under light soaking might be related to the mechanism of trapped charges of the present study, we measured KPFM images on the surface of perovskite after light illumination without ion deposition. As shown in Supplementary Fig. 13, charges are clearly trapped along grain boundaries for the ITO/C$_{60}$/perovskite sample soaked by light, which is in agreement with the previous report[39]. These results confirmed that light soaking alone induces charge trapping along the grain boundaries of perovskite material similar to what was done by introduction of ion charges in the dark condition. From our concept, these trapped charges can now trigger the irreversible degradation due to moisture, as the same happened when ion charges are deposited in the dark. Therefore, the fundamental cause for irreversible degradation would be the same, that is, the trapped charges that could trigger the irreversible degradation under humid air. Such irreversible degradation under light soaking was reported in previous several reports[11,21]; however, the reason has not been clearly elucidated so far, although Christians et al.[11] suggested the lessened hydrogen bonding after photoexcitation as a possible cause. Here we argue that trapped charges under moisture would be responsible for the initiation of irreversible degradation under light or in the dark, as light illumination always generates charges and traps the charges along grain boundaries as shown earlier. This can explain well why moisture itself without illumination or intentional ion deposition only hydrated perovskite reversibly. Light illumination under nitrogen gas without moisture for 2 days was shown to hardly degrade the perovskite as shown in Supplementary Fig. 14, which is strongly in contrast with the case of light illumination under moisture (see Fig. 3j).

## Discussion
We suggest a possible scenario how trapped charge could trigger the irreversible decomposition of perovskite materials. First, in the presence of water molecules, perovskite materials are known

to form hydrates. Within the hydrated perovskite, octahedral PbX$_6^{4-}$ interacts with both organic cations (CH$_3$NH$_3^+$, HC(NH$_2$)$_2^+$) and H$_2$O (ref. 11). Next, the charges trapped at the defect site regardless of polarity could help to deprotonate organic cations by induced local electric field similar to the way that was well known from the soft matter physics on electric-field-induced de-protonation of organic molecules[40,41]. Such de-protonation process[27,38] in the presence of water could yield volatile molecules such as CH$_3$NH$_2$ and HC($=$NH)NH$_2$ that can evaporate at room temperature. The following deprotonation from organic cations could take place due to trapped charge-induced local electric field:

$$\begin{pmatrix} CH_3NH_3^+ (MA^+) \\ HC(NH_2)_2^+ (FA^+) \end{pmatrix} + H_2O \xrightarrow{TC} \begin{pmatrix} CH_3NH_2(\uparrow) \\ HC(=NH)NH_2(\uparrow) \end{pmatrix} + H_3O^+$$
(1)

where TC means trapped charge. Evaporation of resulted volatile neutral molecules could shift the following equilibrium reaction that prevail during the formation of hydrates towards the right-hand side, which causes the beginning of irreversible degradation of perovskite:

$$\begin{pmatrix} CH_3NH_3 \\ HC(NH_2)_2 \end{pmatrix} PbX_3 \overset{H_2O}{\underset{H_2O+TC}{\rightleftharpoons}} PbX_2(s) + \begin{pmatrix} CH_3NH_3^+ \\ HC(NH_2)_2^+ \end{pmatrix}(aq) + X^-(aq) \\ \xrightarrow{} PbX_2(s) + \begin{pmatrix} CH_3NH_2 \\ HC(=NH)NH_2 \end{pmatrix}(\uparrow) + H_3O^+ + X^-(aq)$$
(2)

where X denotes halide. In addition, trapped charges could help the hydration process by distorting the structure of perovskite electrostatically, which leads water molecule to penetrate easily into the perovskite structure.

Next, we investigated the effect of oxygen on the moisture-induced degradation of perovskite materials. Previous studies on device encapsulation have mostly focused on blocking moisture and overlooked the effect of oxygen. However, we found an interesting result that is the synergetic effect of oxygen on the process of moisture-induced degradation of perovskites. To test this, we switched the Gas 2 from N$_2$ to dry air (N$_2$ + O$_2$:8:2) for bubbling water (see Fig. 4a) with the Gas 1 remaining as N$_2$ for ion generation. As shown in Fig. 6b, the addition of oxygen under the same condition of trapped charge and moisture of RH = 40% clearly showed more rapid degradation compared with the case without oxygen (Fig. 6a). We also verified that degradation did not happen if a dry air gas was employed as Gas 2 without moisture (Supplementary Fig. 15). This implies that oxygen alone would not harm the perovskite even under the existence of

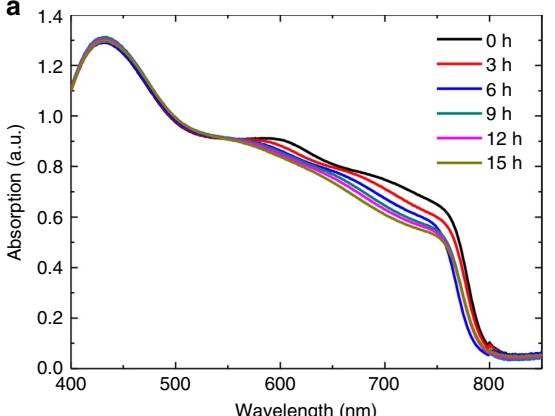
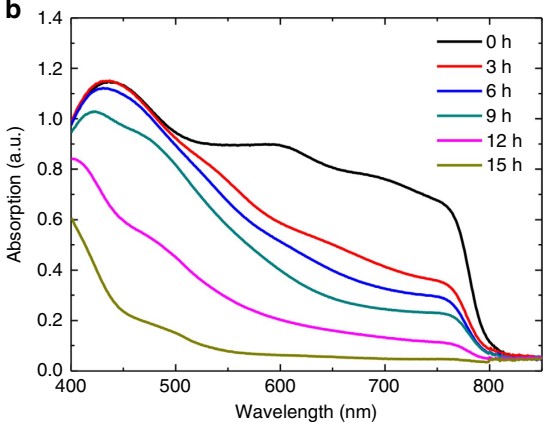

**Figure 6 | Synergetic effect of oxygen on degradation.** (**a,b**) Time evolution of the absorption spectra of MA$_{0.6}$FA$_{0.4}$PbI$_{2.9}$Br$_{0.1}$ films (**a**) in humidified nitrogen and (**b**) in humidified air measured at an interval of 3 h during deposition of positive nitrogen ions. In both cases, the RH is 40%.

trapped charges; however, the oxygen could worsen the degradation process in the presence of water and trapped charges. Previously, Niu *et al.*[42] suggested reaction equations of oxygen-involved degradation in humid air condition, which represent the formation of $H_2O$ as a reaction product. Similarly, we explain the fast degradation in the presence of oxygen more clearly based on the scavenging effect of oxygen on $H_3O^+$ proton generated from irreversible de-protonation process (equation (1)). The process of scavenging $H_3O^+$ by oxygen could be expressed as follows.

$$H_3O^+ + X^- + \frac{1}{4}O_2 \rightarrow \frac{1}{2}X_2 + \frac{3}{2}H_2O \quad (3)$$

The overall chemical reaction of the oxygen involved degradation can be expressed as:

$$\begin{pmatrix} CH_3NH_3 \\ HC(NH_2)_2 \end{pmatrix} PbX_3(s) + \frac{1}{4}O_2 \xrightarrow{H_2O + TC} PbX_2(s)$$
$$+ \begin{pmatrix} CH_3NH_2 \\ HC(=NH)NH_2 \end{pmatrix}(\uparrow) + \frac{1}{2}X_2 + \frac{1}{2}H_2O \quad (4)$$

We also confirmed that solid $I_2$ is produced from the perovskite degraded in ambient air, which is evidenced by XRD results (see Supplementary Fig. 16). Interestingly, the overall reaction produces water, which is in agreement with the previous work[42], and then this water could cause a chain reaction of water-induced degradation. Therefore, oxygen, which comprises about 20% of the atmosphere, should be considered as an additional target that must be avoided together with moisture.

In conclusion, we found that trapped charges would be responsible for triggering the irreversible degradation in the moisture-induced degradation of perovskite materials regardless of charge polarity. To verify this, we designed a novel experimental setup enabling the deposition of different polarity charges on the surface of perovskites for controlled moisture degradation experiments. From this setup, we demonstrated that the perovskite materials degraded irreversibly along grain boundaries only when both moisture and trapped charge exist simultaneously. During the course of study, we developed and used a mixed perovskite material that ensured both high performance and structural stability. Our study explains both why the degradation begins to occur from the different side of interface between perovskite and charge extraction layer for different charge extraction layers and how light soaking always degrades irreversibly in the presence of moisture. KPFM study reveals that charges are trapped preferentially along grain boundaries of perovskites even under uniform deposition of ion charges or uniform illumination of light, which supports our idea of trapped charge-driven degradation. We also found the synergetic effect of oxygen in the process of moisture-induced degradation. The present study suggests that the prevention of accumulation of charges at the interface is very important in addition to proper encapsulation for developing commercially viable perovskite solar cells.

## Methods

**Solar cell fabrication.** ITO glass substrates (AMG, $9.5\,\Omega\,cm^{-2}$, $25 \times 25\,mm^2$) were sequentially sonicated in acetone, isopropanol and deionized water. The cleaned substrate was sufficiently dried in an oven, to eliminate all residual solvents. A 35 nm-thick $C_{60}$ layer[43] was densely coated on the ITO glass substrates by using a vacuum thermal evaporator at the constant rate of $0.1\,\text{Å}\,s^{-1}$. The 50 wt% mixed perovskite solutions (MAI + FAI + MABr : $PbI_2$ : dimethyl sulfoxide = 1:1:1 in dimethylformamide (DMF) solvent) were coated on the ITO/C60 substrate by Lewis base adduct method[25]. To prepare our best compositional solution, 461 mg of $PbI_2$, 79.5 mg of MAI, 68.8 mg of FAI, 11.2 mg of MABr and 78 mg of dimethyl sulfoxide were mixed in 0.55 ml of dimethylformamide (DMF) at room temperature with stirring for 30 min. After spin coating at 4,000 r.p.m. for 20 s with ether dripping treatment, the transparent adduct films were annealed at 130 °C for 20 min, to form black perovskite films. To prepare the HTM, 72.3 mg of Spiro-MeOTAD (Merk) dissolved in 1 ml Chlorobenzen (Sigma-Aldrich) with

28.8 μl of 4-tert-butyl pyridine and 17.5 μl of lithium bis (trifluoromethanesulfonyl) imide solution (520 mg lithium bis (trifluoromethanesulfonyl) imide in 1 ml acetonitrile (Sigma-Aldrich, 99.8%)). The HTM solutions were spin coated onto the perovskite layer at 2,000 r.p.m. for 30 s. After all process, 50 nm gold (Au) as a counter electrode was deposited on the HTM at the rate of $0.3\,\text{Å}\,s^{-1}$ by using a vacuum thermal evaporator. All spin coating processes in our experiments were carried out in a dry room (RH <15%, room temperature).

In the case of $TiO_2$-based devices, a $TiO_2$ blocking layer was fabricated on FTO-coated glass substrates by spin coating 0.15 M titanium di-isopropoxide dis(acetylacetonate) (Sigma-Aldrich, 75 wt% in isopropanol) in 1-butanol (Sigma-Aldrich, 99.8%) at sequentially increasing spin rate of 700 r.p.m. for 8 s, 1,000 r.p.m. for 10 s and 2,000 r.p.m. for 40 s. After spin coating, the $TiO_2$ blocking layer was heated at 125 °C for 5 min and this process was repeated once again. The substrate was annealed at 550 °C for 1 h. The rest of the processes are identical to the fabrication of $C_{60}$-based devices.

**Characterization.** The cross-sectional and surface images of the perovskite films and the fabricated perovskite solar cells were obtained from a high-resolution SEM with a focused ion beam system (Carl Zeiss, AURIGA). The optical absorption spectra of the perovskite films coated on the ITO substrate were measured by ultraviolet-visible spectrophotometer (Agilent Technologies, Cary 5000) in the 400-850 nm wavelength range. The XRD patterns of the perovskite films on the ITO glass were collected by using New D8 Advanced (Bruker) in the $2\theta$ range of 5°–80°. Photocurrent density–voltage curves were measured by a solar simulator (Oriel Sol3A) with Keithley 2400 Source Meter under AM1.5G, which is calibrated to give $100\,mW\,cm^{-2}$ using a standard Si photovoltaic cell (Rc-1000-TC-KG5-N, VLSI Standards). The *J–V* curves were measured by covering devices with a metal mask having an aperture. (6.76 mm²) EQE was measured by a specially designed EQE system (PV measurement Inc.) with 75 W Xenon lamp (USHIO, Japan) as a source of monochromatic light. High-performance long-pass filter (<425 nm, Edmund Optics) was used for aging the devices under ultraviolet-filtered light illumination.

**Data availability.** The data that support the findings of this study are available from the corresponding author upon request.

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

## Acknowledgements

This work was supported by the Global Frontier R&D Program of the Center for Multiscale Energy Systems funded by the National Research Foundation under the Ministry of Education, Science and Technology, Korea (2011-0031561 and 2011-0031577). B.Y.L. acknowledges the support from the National Research Foundation of Korea (NRF) grant funded by the Ministry of Science, ICT and Future Planning (NRF-2016M3A7B4909581). We thank H. Sung and S.R. Noh for discussions and comments on this work.

## Author contributions

N.A., K.K. and M.C. conceived and designed the experiments, and analysed the data results. N.A., K.K., H.Y. and J.B. performed the device fabrication and photovoltaic performance measurements. N.A., K.K. and J.B. carried out the controlled stability test. B.Y.L. and J.-K.L. measured KPFM. P.V.P., N.A., K.K. and M.C. discussed the mechanism. M.C. led the project. N.A., K.K., M.S.J., P.V.P. and M.C. wrote the paper. All authors discussed the results.

## Additional information

**Competing financial interests:** The authors declare no competing financial interests.

