## [Peer Review File · Nature Communications]

Reviewers' comments:

Reviewer #1 (Remarks to the Author):

Authors proposed a plausible mechanism for irreversible degradation of perovskite solar cells under humidity, oxygen and illumination. The manuscript is written well, and the interpretation and conclusion is reasonable from several evidence and analysis. Therefore, there is no serious criticism. The comments found during reviewing this manuscript are below.

1. Authors mentioned in line 167-169 " It is likely that water molecules could penetrate more easily into the distorted tetragonal MAPbI₃ than into the more compact cubic crystal structure of MA_{0.6}FA_{0.4}PbI_{2.9}Br_{0.1}". This is already reported by "Noh, J. H., Im, S. H., Heo J. H., Mandal, T. N. & Seok, S. I. Nano Lett. 13, 1764-1769 (2013)". It would be better to refer previous work.
2. It is stated in line 250 that striking coincidence between two images is the evidence that charges are preferentially trapped along grain boundaries. However, this is in controversy with previous work (J. Phys. Chem. Lett., 2015, 6 (5), pp 875-880). Authors should be discussed with this reference.

Reviewer #2 (Remarks to the Author):

The manuscript presents a thorough study of the degradation of perovskite solar cells in ambient conditions. The authors conclude that upon illumination, trapped charge facilitates a water induced irreversible degradation pathway, which is accelerated by the presence of oxygen. The study presents new insights into the roles played by trapped charge, oxygen, and the electron selective contact used. The observations are interesting and the topic is certainly important to the perovskite solar cell community, as stability appears to be the most pressing concern for this exciting new technology. However, before this manuscript can be considered for publication, there are several technical issues that need to be addressed. The conclusions do not seem to be completely supported by the experimental evidence. Even if these issues are addressed, I probably still would not recommend publication in a Nature journal because it is vastly more important to understand how encapsulated devices degrade than to understand how unencapsulated devices degrade. We don't really need to know in detail how water destroys the device. We know enough to know that the devices must be encapsulated.

1. The authors demonstrate that degradation is accelerated at the TiO₂ contact. This has been demonstrated previously, and has been often ascribed to the presence of deep hole traps on the TiO₂ surface which can photo-oxidize any material it is in contact with. This process tends to be UV initiated, so the authors should demonstrate that their degradation is not observed when the samples are illuminated in UV filtered (< 420 nm) light. On this point, the authors cannot claim that the degradation occurs from the Spiro side of the device based on their SEMs; it appears to be randomly distributed. Moreover, even if it were predominantly at the spiro side, this would not necessarily be because holes are trapped at that interface, it is more likely because the spiro HTL contains the hygroscopic LiTFSI salt and hence will contain a great deal of moisture.
2. The authors could benefit from an improved discussion of their ion induced trapped charge. This is not a commonly used technique in the community, so either an improved explanation or direct evidence that the ions induce charge carrier trapping is required. Can they confirm their topology measurements for purely light induced degradation without the ions?
3. The authors claim that their results are due to trapped charge and not due to the presence of

electric fields, thus attempting to differentiate their results from a recent study that demonstrated the influence of an electric field on the moisture induced degradation. The "non contact" method for applying an electric field should be specified; it is also very likely that this field is predominantly dropped across the air gap in such a setup. Moreover, since the trapped charge induces the degradation by a "field induced deprotonation" reaction, as the authors hypothesize, the conclusions are very similar in that electric fields, whether it be applied or due to trapped charge, induce this irreversible degradation. Can the authors directly monitor the motion of the protons?

4. The authors suggest that I2 and Br2 are two degradation products; can they directly determine if these are evolved? It should be possible to detect the evolution of I2.

5. The authors make a few not fully substantiated claims:

"It is likely that water molecules could penetrate more easily into the distorted tetragonal MAPbI3 than into the more compact cubic crystal structure of MA0.6FA0.4PbI2.9Br0.1." There are many other factors that play into hygroscopicity.

"There was no reason that the degradation should have started from the electrode if electric field alone could cause irreversible degradation" Wouldn't field induced ion motion cause degradation from one side first as ions start to move and deplete the area next to an electrode that doesn't have material and field on the other side to replace the lost ions.

Reviewer #3 (Remarks to the Author):

The Authors investigate the degradation processes in perovskite solar cells, focussing on how the combination of trapped charges, humidity and oxygen can trigger/enhance it. Using a novel setup, they manage to isolate the different factors coming into play, identifying reversible and irreversible effects and synergies. The Authors also provide a reasonable explanation for the degradation process and the different behaviours of perovskite films with different chemical composition or different ETLs.

The work reported here falls into the current research on perovskite cells stability. Previous work in the literature covered some of the degradation factors, but to my knowledge this is the most complete analysis on the subject. In particular, the study of the dynamics of the trapped charges is novel and has very relevant consequences for applications. This study has very significant relevance to the perovskite solar cell community; the findings on charge dynamics will be of interest to a growing community as this class of materials are considered for other electronic applications (such as LEDs).

The experiments are adequately designed and presented; the statistics concerning the cells being examined should be sufficient to support the assumption that the devices being analysed are being representative. The presentation of the work is clear and the conclusions are, in my opinion, sufficiently supported by the experimental data and the suggested model.

Some specific comments:

1) The approach to use C60 as ETL is valuable, but it has been reported in the literature before, so it should be referenced more clearly (for example, DOI: 10.1021/acs.jpcllett.5b00902 J. Phys. Chem. Lett. 2015, 6, 2399–2405).

2) It is known that some labs use air doping to enhance the properties of spiro. This results in the formation of holes/channels in the hole transporting layer, as reported, for example, by Hawash (DOI: 10.1021/cm504022q Chem. Mater. 2015, 27, 562–569). This is not mentioned by the Authors in the cells fabrication, so I assume that in this case there was no air doping step. Could the authors clarify

this, and maybe discuss how the additional porosity would affect cell degradation?

3) Figure 1e: This wasn't clear to me, although it might be a small detail - are the SEM cross-sectional views in panel e taken from the same sample after different times, or were they twin samples? (Same for fig. 3c).

4) Line 219: although the two references reported regarding hydrated perovskites are relevant, I think it would be appropriate to cite the Leguy ChemMat2015 paper on the subject (DOI: 10.1021/acs.chemmater.5b00660 Chem. Mater. 2015, 27, 3397–3407), since it provides good background information.

5) Line 240 - fig. 3c: the Authors state that degradation of the perovskite films starts from the grain boundaries. While I agree that such behaviour should be expected, I am not convinced that the SEM images in fig. 3c prove it. Assuming the SEM top views were acquired from different areas (or different samples), the variations in roughness given by the degradation seem to be considerably larger than the differences in height between grains, so it is hard to understand where the process starts.

6) Line 296 (formulae 1-2): the deprotonation of the organic component of the perovskite films has different chemical routes for MA and FA. Do the Authors think that would influence the overall degradation rate of the film, or would the degradation process limited by the hydration?

7) Figure S5: the degradation in the two cases (C60/TiO₂) seems to cause features of different size (clear in the 36h panel) - the C60 sample has smaller channels (up to 100 nm or so), whereas the TiO₂ sample has significantly larger voids (several hundreds of nm). Is this something that has been observed consistently, and do the Authors have an explanation for it?

Point-By-Point Response to Referees' Comments

Nature Communications Manuscript Revision Request

Manuscript Number: NCOMMS-16-11128-T

Manuscript Type: Article

Title: "Trapped Charge Driven Degradation of Perovskite Solar Cells"

Author(s): Namyong Ahn†, Kwisung Kwak†, Min Seok Jang, Heetae Yoon, Byung Yang Lee, Jong-Kwon Lee, Peter V. Pikhitsa, Junseop Byun, and Mansoo Choi*

First of all, we would like to express our gratitude to all the reviewers for their valuable comments on our manuscript entitled "Trapped Charge Driven Degradation of Perovskite Solar Cells". Herein, we have addressed issues raised from the reviewers and answered these concerns in a separate point-by-point response. We highlighted the revised parts in red in the revised manuscript. As requested by the editor, we also followed format requirements of Nature Communications such as words limit in the abstract, etc.

Reviewer #1 (Remarks to the Author):

Authors proposed a plausible mechanism for irreversible degradation of perovskite solar cells under humidity, oxygen and illumination. The manuscript is written well, and the interpretation and conclusion is reasonable from several evidence and analysis. Therefore, there is no serious criticism. The comments found during reviewing this manuscript are below.

Answer) We thank the reviewer for the encouraging comments

1. Authors mentioned in line 167-169 " It is likely that water molecules could penetrate more easily into the distorted tetragonal MAPbI₃ than into the more compact cubic crystal structure of MA_{0.6}FA_{0.4}PbI_{2.9}Br_{0.1}". This is already reported by "Noh, J. H., Im, S. H.,

Heo J. H., Mandal, T. N. & Seok, S. I. Nano Lett. 13, 1764-1769 (2013)". It would be better to refer previous work.

Answer) We agree that the paper mentioned by the reviewer is relevant to our results. We additionally cited the recommended reference in our revised manuscript.

2.It is stated in line 250 that striking coincidence between two images is the evidence that charges are preferentially trapped along grain boundaries. However, this is in controversy with previous work (J. Phys. Chem. Lett., 2015, 6 (5), pp 875-880). Authors should be discussed with this reference.

Answer) We thank the reviewer for bringing up this paper to us. The paper that the reviewer mentioned also showed Kelvin Probe Force Microscopy (KPFM) images with or without light illumination. In that paper, the perovskite films for KPFM measurement were fabricated on FTO/TiO₂ substrates to quench electrons. Under light illumination, they also showed that charge induced potential distribution was well corresponding to topological grain boundary distribution, that is, high peaks of the potential occurred at low (topographically) grain boundaries. This is consistent with our measurements shown in Figure S13, that is, high peaks of the potential occur at low (topographically) grain boundaries, which is caused by charge trapping preferentially on the grain boundaries. Therefore, the major point on the charge trapping under light illumination is the same as ours.

However, there is one difference to our result, that is, the image for the reference sample without illumination. For their case even without light illumination, they showed potential distributions existed and matched with topography, but, in the opposite way to the light illumination case, that is, now without illumination, high potential peaks occurred at high topographical region and low potential peaks at low topographical region (this is the grain boundary). But in our case, there was no correlation between potential and topography distributions for the case of no illumination. They suggested one possibility to explain why this occurred even without generation of charges (for no illumination case), that is, the possibility of generating built-in potential distribution caused by defects made on the grain boundaries of perovskite film coated on TiO₂ layer. Since our reference sample measured without light illumination (untreated sample shown in Figure S13) has perovskite film just on ITO glass (without charge extraction layer like the TiO₂ of their case), we could think of the

difference in sample structure as the reason of KPFM image difference under the dark. The second possibility would be the cross-talk artefacts between topography and KPFM signals in KPFM imaging, which are well known to be highly dependent on the tip-substrate distance. [Barbet, S. et al., *J. Appl. Phys.* 115, 144313, (2014), Joseph, L. G. et al. *Nanotechnology*. 27, 245705 (2016), Liu, J. et al. *Langmuir*. 31, 10469-10476, (2015).] In case of the paper mentioned by the reviewer, no information could be found in regards to the tip-substrate distance. In our cases, the tip was positioned sufficiently away from the substrate (as far as 20 nm) in order to exclude the cross-talk artefacts from topography. To check the possibility of the sample structure difference, we made additional KPFM experiment under dark using the sample of ITO/C₆₀/Perovskite (now perovskite coated on charge extraction layer like in their case). Note that for our light illumination cases, we used the same ITO/C₆₀/Perovskite structure to extract the electrons because both holes and electrons are generated during light soaking which was described in the supplementary information (Topography and Kelvin probe force microscopy). We measured KPFM images of the ITO/C₆₀/Perovskite sample under dark condition with the tip height being 20 nm, as shown in Figure R1. Even in this case, we could not see clear correlation between topography and surface potential despite the presence of the C₆₀ electron extraction layer. We therefore suspect that KPFM results (for the dark) in the reviewer's reference might have been influenced by the cross-talk artefacts from topography footprints during the measurements. All things considered, we think that our results are not in controversy with the mentioned reference by the reviewer since the major point of KPFM measurements under the illumination is consistent with our results and the difference observed in the dark may be just from different sample structure or measurement condition. We added this discussion in the revised version (page 12 in the revised manuscript and page 6 in the revised supplementary information).

Figure R1. Topography and KPFM images of the ITO/C₆₀/Perovskite sample measured under the dark condition with the tip-substrate distance being 20 nm.

Reviewer #2 (Remarks to the Author):

The manuscript presents a thorough study of the degradation of perovskite solar cells in ambient conditions. The authors conclude that upon illumination, trapped charge facilitates a water induced irreversible degradation pathway, which is accelerated by the presence of oxygen. The study presents new insights into the roles played by trapped charge, oxygen, and the electron selective contact used. The observations are interesting and the topic is certainly important to the perovskite solar cell community, as stability appears to be the most pressing concern for this exciting new technology. However, before this manuscript can be considered for publication, there are several technical issues that need to be addressed. The conclusions do not seem to be completely supported by the experimental evidence. Even if these issues are addressed, I probably still would not recommend publication in a Nature journal because it is vastly more important to understand how encapsulated devices degrade than to understand how unencapsulated devices degrade. We don't really need to know in detail how water destroys the device. We know enough to know that the devices must be encapsulated.

Answer) We thank the reviewer for encouraging comments. We agree that it is important to understand how encapsulated devices degrade. That is why our present work is still of significance, since our work may explain why devices even with encapsulation still degrade. Until now, some factors like moisture, heat and light have been individually known to cause degradation of perovskite materials, however, the truth is that these factors could synthetically influence the degradation. Previous works already revealed that only moisture just causes reversible hydration [Christians, J. A et al. *J. Am. Chem. Soc.* 137, 1530-1538, (2015), Leguy, A. M. A. et al. *Chem. Mater.* 27, 3397-3407, (2015)] and only light also induces reversible change of perovskite. [Gottesman, R. et al. *J. Phys. Chem. Lett.* 6, 2332-2338, (2015)] From our experiments, we demonstrated that the irreversible degradation takes place when both charge and moisture exist simultaneously. In other words, even an extremely small amount of water molecule in the encapsulated device could cause the degradation with trapped charges generated by light. Additionally, oxygen would play a critical role in the degradation of encapsulated devices because oxygen induced degradation could produce

additional water molecules. As the reviewer pointed out, we think that we must encapsulate the device for long-term stability, but encapsulation could not perfectly block moisture and oxygen no matter how tight and strong the encapsulation would be. Considering such a limitation of encapsulation, our work suggests that it can be more important to prevent charges trapped along grain boundaries and interfaces intrinsically for long-term stability of perovskite solar cells.

1. (1) The authors demonstrate that degradation is accelerated at the TiO₂ contact. This has been demonstrated previously, and has been often ascribed to the presence of deep hole traps on the TiO₂ surface which can photo-oxidize any material it is in contact with. This process tends to be UV initiated, so the authors should demonstrate that their degradation is not observed when the samples are illuminated in UV filtered (< 420 nm) light. (2) On this point, the authors cannot claim that the degradation occurs from the Spiro side of the device based on their SEMs; it appears to be randomly distributed. (3) Moreover, even if it were predominantly at the spiro side, this would not necessarily be because holes are trapped at that interface, it is more likely because the spiro HTL contains the hygroscopic LiTFSI salt and hence will contain a great deal of moisture.

Answer) (1) We think that our discussion might give rise to the reviewer's misunderstanding about our SEM results showing different degradation patterns in Figure 1.e,f. Our point is that the degradation would be dominantly initiated at single side interface (ETL or HTL contact) depending on the kinds of charge extraction layer. Although the presence of deep-hole traps or the photocatalytic effect of the TiO₂ layer have previously been suggested as the reasons of the accelerated degradation at the TiO₂ contact, we additionally suggest that charges trapped along the TiO₂/perovskite interface can dominantly cause fast degradation regardless of the UV light induced effect. To verify the effect of UV light on the degradation patterns, we obtained time evolution of the FIB-SEM cross-sectional images of the C₆₀ and TiO₂ based devices aged under UV filtered (Edmund Optics, 425nm High Performance Longpass Filter) light illumination in ambient condition according to the reviewer's comment. As shown in Figure R2, the degradation still occurred for TiO₂ based device even under UV filtered light, which again supports our idea on trapped charge driven degradation (even without photocatalytic effect for UV filtered light, charges could be trapped between TiO₂

and perovskite materials and then could trigger the irreversible degradation as our works demonstrate). It is noted that the SEM images again show different degradation patterns depending on kinds of ETLs in the same manner as for the case without UV filtering in Figure 1.e,f. This means that such degradation patterns always appear regardless of UV light and the starting side of the degradation is determined by types of charge extraction layer. We added this discussion in the revised version. (Supplementary Fig. 6)

Figure R2. Time evolution of the FIB-SEM cross-sectional images of the C₆₀ (left) and TiO₂ (right) based devices employing MA_{0.6}FA_{0.4}PbI_{2.9}Br_{0.1} perovskites, which were aged for 48 h under UV filtered light illumination in ambient conditions. Scale bars = 200 nm.

(2) To investigate whether the degradation is initiated from the Spiro-MeOTAD/Perovskite interface or randomly distributed in the C₆₀-based devices, we measured FIB-SEM cross-sectional images of the C₆₀ based device degraded under light illumination again. As shown in Figure R3, C₆₀-based samples tend to be gradually degraded from the Spiro-MeOTAD side to the C₆₀ side. We added these results in the revised version (Supplementary Fig. 5)

C_{60} -based Device

Figure R3. Time evolution of the FIB-SEM cross-sectional images of the C_{60} based devices employing $MAPbI_3$ perovskites, which were degraded under one sun light illumination in ambient conditions. Scale bars = 200 nm.

(3) As the reviewer pointed out, it could be thought that the degradation would occur from the Spiro-MeOTAD side due to the hygroscopicity of Spiro-MeOTAD. However, we compared C_{60} -based devices and TiO_2 -based devices employing the same HTL(Spiro-MeOTAD) leading to different degradation patterns. This means that the hygroscopicity would not be a dominant factor to determine where the degradation initiates.

2. (1) The authors could benefit from an improved discussion of their ion induced trapped charge. This is not a commonly used technique in the community, so either an improved explanation or direct evidence that the ions induce charge carrier trapping is required. (2) Can

they confirm their topology measurements for purely light induced degradation without the ions?

Answer) (1) We thank the reviewer for the meaningful comments. Ion generation through corona discharge and their movement and deposition under the electric field is well established in the field of Aerosol Science. Furthermore, our group had used corona gas ion generation and deposition for patterning metal nanoparticles in the previous works. [Kim, H. et al. *Nat. Nanotechnol.* 1, 117-121, (2006), Lee, H. et al. *Nano Lett.* 11, 119-124, (2011)] This technique is called ion-assisted aerosol lithography (IAAL) which has been developed in our research group for depositing charged nanoparticles at specific location by inducing electrostatic lens caused by accumulated corona gas ions on the patterned photoresist (PR) as shown in Figure R5. In the papers, we showed KPFM measured potential distributions caused by ion deposition on the patterned sample as shown in Figure R4. This is the direct evidence that ions are deposited on the patterns under the given electric field. In this present work, we used the same method for ion charge deposition on the perovskite surface and confirmed charge trapping by measuring KPFM images. Since a detailed explanation of corona ion induced charge trapping is still required, as the reviewer pointed out, we additionally cited two previous reports published by our group to clearly explain the experiments and a text book for an overall understanding about ion generation and transport.[Hinds, C. W. *Aerosol Technology : properties, behavior, and measurement of airborne particles. A Wiley-Interscience Publication* 2, 331-332, (1999)]

Answer) (2) We measured KPFM images to provide a direct evidence of ion-induced charge trapping at the grain boundaries as can be seen in Figure S13. We assure that the samples for purely light induce degradation was made without ions and topology measurements for those samples were done without ions.

Figure R4. AFM image of patterned sample (a) and (b,c,d) KPFM image showing potential distribution as a function of ion flow rate (Ion flow rate: (b) = 3 L min⁻¹ (c) = 4 L min⁻¹ (d) = 6 L min⁻¹) [Kim, H. *et al. Nat Nanotechnol.* **1**, 117-121, (2006)]

Figure R5. Illustration of ion-assisted aerosol lithography : red balls sketched on the surface of photoresist(PR) are ions deposited and green arrows show repulsion field to the same polarity charged nanoparticles due to accumulated ions and yellow arrows show attractive

field to the substrate. Both of repulsive and attractive field are combined to generate electrostatic lens through which charged nanoparticles are focused.

[Lee, H. *et al. Nano Lett.* **11**, 119-124, (2011)]

3. (1) The authors claim that their results are due to trapped charge and not due to the presence of electric fields, thus attempting to differentiate their results from a recent study that demonstrated the influence of an electric field on the moisture induced degradation. The "non contact" method for applying an electric field should be specified; it is also very likely that this field is predominantly dropped across the air gap in such a setup. Moreover, since the trapped charge induces the degradation by a "field induced deprotonation" reaction, as the authors hypothesize, the conclusions are very similar in that electric fields, whether it be applied or due to trapped charge, induce this irreversible degradation. (2) Can the authors directly monitor the motion of the protons?

Answer) (1) Leijtens et al. found the irreversible degradation near the gold electrodes coated on perovskite film by applying a weak external field of 600 V/cm in the presence of moisture and attributed it to the ion movement through electric field. [Leijtens, T. et al. *Advanced Energy Materials* 5, 1500962, (2015)] Since an electric field was applied between two electrodes touching perovskite film in their experiment, electric current could flow and there was a possibility of charge trapping underneath the electrode, which might have played a role for degradation. This means that we can not definitely conclude that the observed degradation would have been caused by given external electric field. There is a possibility that degradation could have been initiated by charges trapped underneath the electrode. To isolate the effect of pure external electric field, we examined the degradation of perovskite materials by applying non-contact electric field which was given by two floating electrodes; one electrode is in air above perovskite film coated on ITO glass and the other electrode exists beneath the glass. We found no degradation up to 12 kV/cm under 90 % RH. We agree with the reviewer that this field will be dropped across the air gap and therefore, the real field inside perovskite film should be different from the given field. Note also that the perovskite film might be uniformly polarized by one-directional strong E-field because perovskite materials are known to have a high dielectric constant [Lin, Q. et al. *Nature Photon.* 9, 106-112, (2015), Juarez-Perez, E.J. et al. *J. Phys. Chem. Lett.* 2, 2390-2394, (2014)]. Therefore, further study should be needed to completely understand the effect of pure external electric field.

It is noted that the differences between the fields due to the trapped charges and the external field lie in the point-like character of the trapped charges, which produce locally huge and irregular fields. The huge and irregular fields formed by charges trapped along grain boundaries could help the process of deprotonation .

We discussed about this in the revision. (page 11)

(2) Currently, we cannot directly monitor the motion of the protons, however, we are performing quantum calculation to dig into further detailed pathways.

4. The authors suggest that I₂ and Br₂ are two degradation products; can they directly determine if these are evolved? It should be possible to detect the evolution of I₂.

Answer) Niu et al. already detected the evolution of I₂ in the previous work, which was evidenced by X-ray diffraction. [Niu, G. et al. *J. Mater. Chem. A* 2, 705-710, (2014)] Our results also show that the peak originating from I₂ appears after aging for 10hr in ambient air under one sun illumination as shown in Figure R6. We added this figure in our revised supplementary information as Figure S16.

Figure R6. X-ray diffraction (XRD) patterns of the fresh and degraded samples of MAPbI₃ and MA_{0.6}FA_{0.4}PbI_{2.9}Br_{0.1}. The graph shows the magnified XRD patterns around the peak originating from I₂.

5. The authors make a few not fully substantiated claims:

"It is likely that water molecules could penetrate more easily into the distorted tetragonal MAPbI₃ than into the more compact cubic crystal structure of MA_{0.6}FA_{0.4}PbI_{2.9}Br_{0.1}."

There are many other factors that play into hydroscopicity.

"There was no reason that the degradation should have started from the electrode if electric field alone could cause irreversible degradation" Wouldn't field induced ion motion cause degradation from one side first as ions start to move and deplete the area next to an electrode that doesn't have material and field on the other side to replace the lost ions.

Answer) We think that the former claim may be sufficiently substantiated from our experiments. And also another researcher also suggested this previously as pointed out by the Reviewer #1 [Noh et al. *Nano Lett.* 13, 1764-1769 (2013)] . We already compared the value of tolerance factor (Figure S1), stability under light illumination (Figure S2), and the hydration of both MAPbI₃ and MA_{0.6}FA_{0.4}PbI_{2.9}Br_{0.1} to explain that our new mixed perovskites, MA_{0.6}FA_{0.4}PbI_{2.9}Br_{0.1}, have the most compact cubic crystal structure. Furthermore, we think that the penetration of water molecules, in this case, would be dominantly determined by the density of the different crystal structures, although there are many other factors to determine the hygroscopicity. For the second part of the statement on the electric field effect, we agree that the statement has not been fully proved and further study should be needed to clarify the effect of pure (macroscopic) electric field on the degradation as mentioned above. Therefore, in the revised version, we eliminate this statement. Instead, we mentioned a possibility of degradation that could be caused by trapped charges under the gold electrode. "there was a possibility of charge trapping underneath the electrode, which might have played a role for degradation." (page 10 in the revised version).

Reviewer #3 (Remarks to the Author):

The Authors investigate the degradation processes in perovskite solar cells, focussing on how the combination of trapped charges, humidity and oxygen can trigger/enhance it. Using a novel setup, they manage to isolate the different factors coming into play, identifying reversible and irreversible effects and synergies. The Authors also provide a reasonable explanation for the degradation process and the different behaviours of perovskite films with different chemical composition or different ETLs.

The work reported here falls into the current research on perovskite cells stability. Previous work in the literature covered some of the degradation factors, but to my knowledge this is the most complete analysis on the subject. In particular, the study of the dynamics of the trapped charges is novel and has very relevant consequences for applications. This study has very significant relevance to the perovskite solar cell community; the findings on charge dynamics will be of interest to a growing community as this class of materials are considered for other electronic applications (such as LEDs).

The experiments are adequately designed and presented; the statistics concerning the cells being examined should be sufficient to support the assumption that the devices being analysed are being representative. The presentation of the work is clear and the conclusions are, in my opinion, sufficiently supported by the experimental data and the suggested model.

Answer) We thank the reviewer for the kind and encouraging comments on our present work.

Some specific comments:

1) The approach to use C60 as ETL is valuable, but it has been reported in the literature before, so it should be referenced more clearly (for example, DOI: 10.1021/acs.jpcelett.5b00902 J. Phys. Chem. Lett. 2015, 6, 2399–2405).

Answer) We have added the recommended reference in our revised manuscript.

2) It is known that some labs use air doping to enhance the properties of spiro. This results in the formation of holes/channels in the hole transporting layer, as reported, for example, by Hawash (DOI: 10.1021/cm504022q Chem. Mater. 2015, 27, 562–569). This is not mentioned by the Authors in the cells fabrication, so I assume that in this case there was no air doping step. Could the authors clarify this, and maybe discuss how the additional porosity would affect cell degradation?

Answer) There was no additional air doping step of Spiro-MeOTAD during the cell fabrication. But, in our experiments, the Spiro-MeOTAD layer would be slightly doped by air because we fabricated the layer by spin-coating in ambient air. We investigated the morphology of the Spiro-MeOTAD layer by SEM measurements as shown below. In our case, there were no obvious pinholes unlike the argument claimed by Hawash. The Spiro-MeOTAD layer shows the full coverage of the perovskite as can be seen in cross-sectional images of the device. Some hole-like patterns appearing at the top-view images would be due to just morphological roughness, which are not due to the presence of the pinholes. If there are real pinholes that can lead to direct exposure of the perovskite layer to air, the cell will absolutely be degraded faster than the case of no pinholes. Note that the same Spiro-MeOTAD was used for the cells using different electron extraction layers, which showed opposite degradation behavior as discussed in our paper.

Spiro-MeOTAD Morphology

Figure R7. The top-view and cross-sectional SEM images to show the Spiro-MeOTAD morphology.

3) Figure 1e: This wasn't clear to me, although it might be a small detail - are the SEM cross-sectional views in panel e taken from the same sample after different times, or were they twin samples? (Same for fig. 3c).

Answer) The images in Fig. 1e and Fig. 3c were obtained from different samples, but all the samples were twin samples prepared by the same fabrication process and aged for different hours. Accordingly, there are some morphological differences between each sample.

4) Line 219: although the two references reported regarding hydrated perovskites are relevant, I think it would be appropriate to cite the Leguy ChemMat2015 paper on the subject (DOI: 10.1021/acs.chemmater.5b00660 Chem. Mater. 2015, 27, 3397–3407), since it provides good background information.

Answer) The relevant paper reported by Leguy et al. was already cited in the original manuscript as 16th reference. (Line 412) We additionally cited the reference at the statement corresponding to line 219.

5) Line 240 - fig. 3c: the Authors state that degradation of the perovskite films starts from the grain boundaries. While I agree that such behaviour should be expected, I am not convinced that the SEM images in fig. 3c prove it. Assuming the SEM top views were acquired from different areas (or different samples), the variations in roughness given by the degradation seem to be considerably larger than the differences in height between grains, so it is hard to understand where the process starts.

Answer) All the SEM images in Fig. 3c were measured at a tilted angle in order to obtain cross-sectional views. To more clearly show that the degradation of perovskite films would start at the grain boundaries, we again measured the top-view images without tilting as shown in Figure R8. The images were obtained from the two same samples, which were degraded under light illumination or ion charge deposition for 9 hours, respectively. These top-view images show deepening cracks between grains formed by degradation. Since we could not find the exact same area for each measurement of SEM, it is true that the SEM images could not show the degradation on the exact same area. Though, based on the examination of degradation patterns it looks like that such cracks lie on grain boundaries. The reason why the roughness made by the degradation is different from the size of the grains would be because degradation could not start from all the grain boundaries at the same time. As the below figure shows, degradation could begin from certain grain boundaries possibly having more defects. Therefore, the degradation pattern scale could be larger than the grain size scale. We discussed this in the revision (page 11) and added top view image as Supplementary Fig. 11.

Figure R8. The top-view SEM images of the fresh and degraded perovskite film. The degraded samples were aged for 9 hours under one sun light illumination (first row) and ion charge deposition (second row).

6) Line 296 (formulae 1-2): the deprotonation of the organic component of the perovskite films has different chemical routes for MA and FA. Do the Authors think that would influence the overall degradation rate of the film, or would the degradation process limited by the hydration?

Answer) As the reviewer pointed out, it is likely that the chemical characteristics of the organic cations would influence the overall degradation. Although we cannot trace the chemical routes for deprotonation of different perovskite materials in detail, the chemical properties of the products of the deprotonation can be helpful to anticipate the influence on the overall degradation rate. For example, the boiling point of CH_3NH_2 (the product of the deprotonation of MA cation) is lower than that of $\text{HC}(\text{NH})(\text{NH}_2)$ (the product of the deprotonation of FA cation), which indicates that the deprotonation of MA can be faster than that of FA due to rapid evaporation of CH_3NH_2 according to law of le Chatelier.

Nevertheless, we still believe that the crystal structure of the perovskite determined by ionic radii would be more dominant to the degradation rate, which is evidenced by our results that show the enhanced stability of our mixed perovskite, $\text{MA}_{0.6}\text{FA}_{0.4}\text{PbI}_{2.9}\text{Br}_{0.1}$, in terms of hydration and light soaking as shown in Figure 2. If chemical routes of organic components do influence the degradation more dominantly than the crystal structure, the degradation rate

of our mixed perovskite ($\text{MA}_{0.6}\text{FA}_{0.4}\text{PbI}_{2.9}\text{Br}_{0.1}$) will be similar to that of MAPbI_3 until all of MA components disappear. Accordingly, the degradation rate may be mostly related in the hydration state depending on crystal structure even if the degradation process is synthetically influenced by various factors such as different chemical routes

7) Figure S5: the degradation in the two cases (C60/TiO2) seems to cause features of different size (clear in the 36h panel) - the C60 sample has smaller channels (up to 100 nm or so), whereas the TiO2 sample has significantly larger voids (several hundreds of nm). Is this something that has been observed consistently, and do the Authors have an explanation for it?

Answer) We have consistently observed that the TiO_2 samples have larger voids compared to the C_{60} based samples. We assert that this difference in void size originates from faster degradation of the TiO_2 samples: As the perovskite is gradually degraded, neighboring microvoids are merged to form larger voids, and thus the size of voids presents the progression of the degradation. As we discussed in the text, C_{60} and TiO_2 are very different in respect of the trapped charge: TiO_2 case has considerable hysteresis that characterizes the trapped charges and charge accumulation. As far as the trapped charges would be the driving force for irreversible degradation, the larger void (or faster degradation) for TiO_2 case can be expected.

REVIEWERS' COMMENTS:

Reviewer #1 (Remarks to the Author):

The authors addressed comments properly. This reviewer has no more criticism with revised manuscript.

Reviewer #2 (Remarks to the Author):

The authors have done a great job of responding to all of the reviewer comments and I think that the revised manuscript should be published.

Reviewer #3 (Remarks to the Author):

The authors adequately addressed the points raised in my first round, and I think the paper is now more convincing.

I suspect that inserting some of the comments from the answer to the referees into the main body of the paper might make things clearer, if that doesn't end up making the manuscript too long. More specifically, I'd mention the brief exposure to air during spin-coating of the cells, and maybe a brief summary of the answer to my point 6) - I'll leave whether to have those in or not to the Authors and Editor.

Point-By-Point Response to Referees' Comments

Nature Communications Manuscript Revision Request

Manuscript Number: NCOMMS-16-11128A

Manuscript Type: Article

Title: "Trapped Charge Driven Degradation of Perovskite Solar Cells"

Author(s): Namyoung Ahn[†], Kwisung Kwak[†], Min Seok Jang, Heetae Yoon, Byung Yang Lee, Jong-Kwon Lee, Peter V. Pikhitsa, Junseop Byun, and Mansoo Choi*

Reviewer #1 (Remarks to the Author):

The authors addressed comments properly. This reviewer has no more criticism with revised manuscript.

Answer) We thank the reviewer for the comment.

Reviewer #2 (Remarks to the Author):

The authors have done a great job of responding to all of the reviewer comments and I think that the revised manuscript should be published.

Answer) We thank the reviewer for the comment.

Reviewer #3 (Remarks to the Author):

The authors adequately addressed the points raised in my first round, and I think the paper is now more convincing. I suspect that inserting some of the comments from the answer to the referees into the main body of the paper might make things clearer, if that doesn't end up making the manuscript too long. More specifically, I'd mention the brief exposure to air during spin-coating of the cells, and maybe a brief summary of the answer to my point 6) - I'll leave whether to have those in or not to the Authors and Editor.

Answer) We thank the reviewer for the comments. According to the reviewer's comment, we added this sentence, "All spin-coating processes in our experiments were carried out in a dry room" in methods section of our revised manuscript. Since our answer to reviewer's point 6 will be open as it is and the conclusion of this discussion would need further experiments, we better leave it in the open peer review file for the reviewer's comments and our answers.